# A method based on 3D affine alignment for the quantification of palatal expansion

**Andrea Maggiordomo[1]☯, Marco Farronato[2]☯ \*, Gianluca Tartaglia[2,3,4], Marco Tarini[1]**

**1** Department of Computer Science, Università Degli Studi di Milano, Milano, Italy, **2** Department of Medicine, Surgery, and Dentistry, Università Degli Studi di Milano, Milano, Italy, **3** UOC Maxillo-facial surgery and dentistry Fondazione IRCCS Cà Granda, Ospedale Maggiore Policlinico, Milan, Italy, **4** Department of Orthodontics, School of Dentistry, University of Milan, Milan, Italy

☯ These authors contributed equally to this work.
\* marco.farronato@unimi.it

**Data Availability Statement:** All relevant data are within the paper and its Supporting Information files.

**Funding:** The authors received no specific funding for this work.

## Abstract

### Introduction

The current methodologies to quantify the palatal expansion are based on a preliminary rigid superimposition of 3D digital models representing the status of a given patient at different times. A new method based on affine alignment is proposed and compared to the gold standard, leading to the automatic analysis of 3-dimensional structural changes and to a simple numeric quantification of overall expansion vector and a better alignment of the digital models.

### Materials and methods

40 digital models (timing span delta 25.8 ± 12.5 months) from young patients (mean age 10.7 ± 2.6) treated with two different palatal expansion techniques (20 subjects with RME—Rapid Maxillary Expander, and 20 subjects with NiTiSE, NiTi self-expander) were superimposed with the new affine alignment technique implemented as an extension package of the open-source MeshLab, from a golden standard starting point of rigid alignment. The results were then compared.

### Results

The new measurement function indicates a mean expansion expressed in a single numeric value of 9.3%, 10.3% for the RME group and 8.4% for the NiTiSE group respectively. The comparison with the golden standard showed a decrease to the average error from 0.91 mm to 0.58 mm.

### Conclusions

Affine alignment improves the current perspective of structural change quantification in the specific group of growing patients treated with palatal expanders giving the clinician useful information on the 3-dimensional morphological changes.

**Competing interests:** The authors have declared that no competing interests exist.

## Introduction

With the recent exploit of 3D images, the digitization of anatomical structures rapidly became the main source of anthropometric data. This technological advancement has important consequences in the dentistry field, unlocking the possibility for dental scientists and practitioners to accurately measure the impact of their therapies, in terms of extent and direction. The improvement in biometry accuracy is crucial to plan surgical, orthopedic, and/or orthodontic-related therapies.

The 3D scans allows to aggregate into a single numerical assessment a large multitude of measurements differently from the 2D linear measurements performed over captured images or even the direct 3D measurements performed on the patient or on physical models. For example, even a low-resolution range scanned model is made by a group of tens of thousands metric measurements, while traditionally the palatal expansion values were obtained by linear distances measured between fixed dental points.

A paradigm for the use of 3D images in clinical practice is given: the capture of a 3D polygonal mesh that faithfully models the physical configuration of a given patient at a given time. The digitization process can be implemented leveraging a variety of different technologies, such as x-rays machines followed by isosurface extraction, range scanning (e.g., laser scanning), which can be performed either on a plaster cast or directly on the patient by the use of IOS (intra oral scanners), image-based techniques such as stereophotogrammetry, and others. There are important differences in terms of costs, required equipment, accuracy, acquisition time, automatism, but the technologies used for the image acquisition are substantially interchangeable. Their output is a polygonal mesh that can be analyzed using geometry processing methodologies and algorithms, to ultimately extract the data. Normally, the acquisition process is repeated for the same patient at different stages of the therapy, during the natural history of a pathology, or throughout growth, development, and aging, and the analysis ultimately focuses on the comparisons on the resulting 3D models.

In this work, we investigate the application of a novel technology to quantify size and shape modifications of dental arches and related structures after the orthodontic/orthopedic treatment. In complex systems, like the maxillo-mandibular with orthodontic and orthopedic procedures, the palatal expansion can lead to shape deformation of the anatomical structures with different vectors and variation entity, which can not be quantified without a fully three-dimensional analysis. The premise of our work is that, in this context, *affine transformations* offer a sufficiently accurate model for the deformation, making it the ideal analytical tool to extract intuitive measures of the undergoing three-dimensional deformation. Namely, we can exploit the singular values decomposition (SVD) to extract from the linear portion of the affine transformation the three orthogonal directions of the expansion and their corresponding expansion coefficients. We argue that these values could provide a high-level geometric description of shape changes, easily conveyable even by non-expert users, yielding more insightful and accurate analysis by effortless automatic procedures. The automatization is particularly useful to make a system available to the clinicians which are not familiar with computing sciences.

Indeed, understanding and predicting the clinical treatment effect is the ultimate goal of decades of orthodontic research [1]. The geometrical morphometric analysis (GMA), which is the science behind the affine transformation, starts with the superimposition of two similar 3D meshes, and is a well-described and solid task that can be used to achieve the above-mentioned goal [2]. Normally, the initial superimposition can be obtained through automatic ICP (Iterative Closest Points) or semi-automatic procedures (BFA; best fit alignment of manually selected paired points) over the six degrees of freedom of a rigid roto-translation [3].

Pre- and post-treatment superimposition can be easily done in any case of intervention to a single or a group of dental elements. The anatomical or dental elements [4] which were not affected by the treatment can be used as reference points [4, 5], as well as artificial objects used in the treatment (e.g. dental implants or miniscrews) [6, 7] which clinicians speculate are not subject to movement during treatment or growth. Notwithstanding, this approach has different limits in orthodontics and maxillofacial surgery because of the intrinsic nature of the spatial deformation during growth or other clinical treatments, like the palatal expansion. In this case, the identified structures cannot be used as stable landmarks through different clinical treatments.

To solve these limitations, clinicians started to investigate which anatomical structures can be considered as so stable to be used as reference points. Recently, the use of hard palate structures like palatal rugae [8] has reached clinical consideration. Palatal rugae can be used as a visual reference by the operator to place reference points for the best fit alignment or as Region of Interest (ROI). Unfortunately, the reference point identification is operator-dependent and using this approach the superimposition might be even more complex, the method error would be high and the final quantification of the treatment effect could be erroneous.

Furthermore, to better detect and understand the shape modifications not influenced by size growth and growth directions during treatment or different treatments, the GMA superimpositions systems need to be taken into consideration. Indeed, size variations can confuse and hide the real effect of mechanical or surgical treatment. Other methodologies described in the literature, as the RFD superimposition, overcome this problems suggesting an initial superimposition of the stable palatal area, and then perform iterations between the occlusal surfaces by the use of ICProx (iterative closest proximity) algorithms by analyzing their final relative position. (cite)

The method for the quantification of the dimensional stress carried by a treatment to an anatomical structure is the procedure that usually follows the superimposition of digital models (DM). This can be calculated by the mean square distance or the root mean square (RMS) distance between the two meshes. Indeed, data resulting from this procedure do not consider important factors as craniofacial growth, dental eruption, and non-regular arch development. This specific aspect is intrinsic to the iteration algorithm used between two models which have changed radically due to the effects of maturation, dental exfoliation and eruption, and treatment. The alignment procedure is called rigid alignment since no modifications to the initial models were performed as a roto-translation occurs to non-fixed mesh. By this operation the difference between the unchanged structures of the palate and the changing structure of the dental arch is resumed under the same number, underestimating the treatment and development values, leading to Anisotropic and Inhomogeneous errors [9].

In contrast, the use of affine (e.g. non-rigid) registrations may provide a better analysis of the actual effect of shape variations independently from size modifications. The first uses of non-rigid registrations in medical imaging have been in the fields of cerebral [10] and breast imaging [11, 12]. The first use for oral imaging was proposed by Leung et al [13]. Non-rigid alignment provides affine linear transformation so that a target shape aligns perfectly to the reference starting from a rough alignment (Fig 1).

The present research aims to propose a new method for non-rigid alignment of 3D digital reproductions of dental arches and related structures to precisely evaluate dimensions and shape changes between two meshes taken, for example, at different time frames by a simple numeric output, expressed as a percentage, and by a vector. To evaluate the new method we propose as a primary outcome the measurements and comparisons between the traditional alignment error, which is considered the golden standard, and the affine automatic measures,

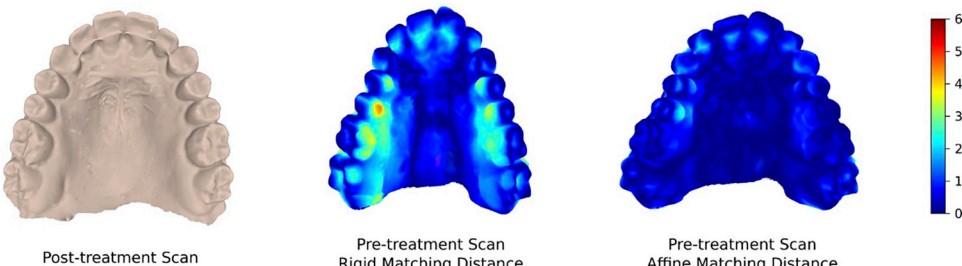

**Fig 1.** Comparison of affine (c) and rigid (b) registration errors (increasing scale towards red) when superimposed to the post-treatment model (a). Affine registration accurately models the palatal deformation producing a noticeably lower alignment error of the scans pre and post-treatment (showing uniform blue coloration). Pre-treatment rigid shows how the rigid alignment show allows to calculate the distance on the palatal side of the premolars, where the distance is higher but fails to measure any difference in the vestibular side, instead, affine alignment correctly stretches the two inputs resulting in a correct overlay.

in two different palatal expansion techniques. The starting point of affine alignment is the golden standard.

## Method

The input of our method is a pair of polygonal meshes $M_0$, $M_1$ representing two different time frames of a given patient, each modeling the same section of dental arch or mandibular parts. This input must be prepared by selecting the appropriate parts of the scanned data. The output of our method consists of the following high-level geometric information extracted from the data:

1. An estimation $\sigma_0$ of the relative expansion (strain) that occurred in between the two inputs

2. The axis where this expansion occurred in the model $M_0$ and $M_1$

### Traditional solutions, based on rigid alignment

The traditional approach consists of performing an initial rigid alignment between $M_0$ and $M_1$, which can be described in identifying the rigid motion (a rotation followed by a translation) $R$ that maps $M_1$ into $M_0$ with the minimal discrepancy. Once $R$ is identified, the models $M_0$ and $R(M_1)$ must be compared, for example, measuring the distance between certain key locations. In this framework, the transformation $R$ serves solely to put $M_0$ and $M_1$ in ideally the same reference frame, counteracting the fact that either model is captured independently, and is recovered in its reference frame.

### Our solution

Instead of a rigid transformation $R$, in our framework, we seek for an affine transformation $A$ mapping $M_1$ into $M_0$ with the minimal residual discrepancy. Affine transformations are a superclass of rigid transformations that, in addition to rotation and translation, include non-rigid deformations such as anisotropic scaling and shearing. The subproblem of finding the optimal transformation $A$ is analogous to the process of finding the rigid transformation $R$ of the traditional pipeline and presents similar challenges and solutions. Then, the sought high-level information is extracted directly from $A$, bypassing the need of any further processing on $M_0$ or $M_1$.

## Rationale

From an algorithmic point of view, the proposed solution is only a minor modification of the traditional ICP based pipeline, but the benefits are substantial:

- *Better alignment*: affine transformations are a wider class of transformations that can account better for the shape differences between $M_0$ and $M_1$ compared to rigid transformations. In addition to being able to account for the unmatching reference frames of $M_0$ and $M_1$ it is able to account, to a first order approximation, the actual physical deformation that the patient underwent between the two captures. Moreover, since we are able to express better alignments with smaller residual errors the ICP algorithm converges more reliably to the target shape, and with a wider convergence basis. See for example Fig 1.

- *Extraction of aggregate data*: in the traditional approach the rigid alignment is only preliminary to the processing and the extraction of the clinically relevant data; in contrast, our affine transformation already captures the sought data. Descending from (a variation of) the ICP algorithm, this data implicitly aggregates a large multitude of captured data samples, instead of only the small subset used in the subsequent measurements. In addition, these data are at the same time concise and descriptive, coming with a straightforward geometric interpretation.

## Step 0: Data preparation

The range scanned data must be cropped to leave the relevant parts only. Part of the range scans $M_0$ and $M_1$ which will be used by our modified ICP must be a linear (i.e. affine) transformation of each other. To this end, parts that are external to the mandibular or cranial data must be removed from the two polygonal meshes. If the range scans have been performed on plaster casts, the bases must also be removed.

## Step 1: Extraction of the affine alignment

Given the two surfaces $M_0$ and $M_1$, we seek the affine transformation $A$ that minimizes the squared geometric discrepancy between $M_0$ and $A(M_1)$. As customary, $A$ is internally represented as a 4×4 matrix with the last row set as the identity.

Just as in the rigid transformation case, we split this task into a user-assisted coarse alignment and a fully automatic fine alignment. The coarse alignment phase is the same as in the rigid case: in our method, we employ a standard point selection method: namely, a user manually selects a set of point correspondences by picking locations in $M_0$ and $M_1$, and the system produces the initial alignment matrix $A_0$ that best matches the pairs. The only difference is that, because an affine matrix has 12 degrees of freedom (versus the 6 degrees of freedom of a rigid transformation), we need the user to identify a minimum of 4 (non-coplanar) points and more for reliability. In our experiments, we used at least 10 points located as follows: 2 points for each side located along the first palatal rugae most evident morphologic features and the last point alongside the palatal midline or the palatal incisal papilla.

The fine alignment phase is performed with a close adaptation of ICP. At each iteration $k$, we identify a suitable subset of point-to-point correspondences as a subset of $n$ point pairs in $M_0$ and $M_1$ presenting a small Euclidean distance between $M_0$ and $M_1' = A_k(M_1)$. Then, $A_{k+1}$ is found as the affine matrix which minimizes the discrepancy between $M_0$ and $A_{k+1}(M_1)$, over the selected $n$ points. Just as in the case of the rigid ICP, we use a spatial indexing structure to quickly identify pairs of close points in $M_0$ and $M_1'$. The iterations are repeated until

convergence. Just as in the standard ICP the Fiducial Registration Error (FRE) can be used as a measure of success of the alignment phase. FRE is defined as the RMS distance between corresponding points in the last iteration. An unsuccessful alignment indicates that the coarse alignment was not sufficiently accurate and must be repeated.

**Solving for matrix A.** Both the coarse and the fine stages require the identification of the best affine matrix $A$ that brings a given set of positions $p_1, \ldots, p_n \in M_1$ into a set of matching positions $q_1, \ldots, q_n \in M_0$.

In other words, we need to solve for

$$A = argmin_{B \in \mathbb{R}^{4 \times 4}} \sum_i |\boldsymbol{p}_i - B(\boldsymbol{q}_i)|_2^2. \tag{1}$$

We write the upper mart of matrix $A$ as $(A_{3 \times 3} | \boldsymbol{t}_A)$. Its translational part $t_a \in \mathbb{R}^3$ can be computed in closed form simply as the difference between the two barycenters of the two sets of points:

$$\boldsymbol{t}_A = \overline{\boldsymbol{p}} - \overline{\boldsymbol{q}}, \tag{2}$$

where $\overline{\boldsymbol{p}}$ and $\overline{\boldsymbol{q}}$ are the barycenters of the two respective point sets (i.e. their mean). Eq (2) descends from (1), because of the linear nature of the transformation $A$ (just as it the case for rigid transformations).

The diagonal submatrix $A_{3 \times 3}$ of $A$ can then be found by solving

$$A_{3 \times 3} = argmin_{B \in \mathbb{R}^{3 \times 3}} \sum_i |(\boldsymbol{p}_i - \overline{\boldsymbol{p}}) - B(\boldsymbol{q}_i - \overline{\boldsymbol{q}})|_2^2. \tag{3}$$

The minimizer can be found using a simple linear least squares system with 9 variables, one for each element of $A_{3 \times 3}$, and $3n$ equations, one for each coordinate of a point pair $(\boldsymbol{p}_i, \boldsymbol{q}_i)$.

## Step 2: Analysis of the affine alignment

Once we have the non-rigid affine matrix $A$, we analyse it to extract the sought aggregate data. We extract its 3×3 submatrix $A^{(3)}$, discarding the last column of $A$, i.e. its translation part, which only represents the difference in reference frames and bears no clinical relevance. Then, we proceed to compute the SVD of $A$

$$A = USV^T = \begin{pmatrix} \vdots & \vdots & \vdots \\ \boldsymbol{u}_0 & \boldsymbol{u}_1 & \boldsymbol{u}_2 \\ \vdots & \vdots & \vdots \end{pmatrix} \begin{pmatrix} \sigma_0 & & \\ & \sigma_1 & \\ & & \sigma_2 \end{pmatrix} \begin{pmatrix} \vdots & \vdots & \vdots \\ \boldsymbol{v}_0 & \boldsymbol{v}_1 & \boldsymbol{v}_2 \\ \vdots & \vdots & \vdots \end{pmatrix}^T \tag{4}$$

with $\sigma_0 \geq \sigma_1 \geq \sigma_2$ non-negative scalars, and $U$ and $V$ orthogonal matrices.

This procedure can be understood as the process to distill the anisotropic scaling of $A$, expressed by the singular values $\sigma_0$, $\sigma_1$, $\sigma_2$, which has clinical relevance, from its rotational part $(UV^T)$ and its translational part $\boldsymbol{t}_A$, which solely reflect the arbitrary choice of the reference frames in which $M_0$ and $M_1$ happened to be expressed, and has no clinical relevance. It bears to stress the fundamental differences that exist between our use of SVD and its use in the standard ICP alignment algorithm (in spite of a superficial similarity). First, in the standard ICP the SVD is applied to the covariance matrix computed from the point pairs, whereas, in our case, it is applied to the matrix found by minimizing the summed squared discrepancies. Second, in the standard ICP, this process is performed at every step, to ensure that only rotation matrices are used, whereas in our case this it is only performed on the final matrix, after

convergence, allowing for affine but not necessarily rigid transformations during the iterative process, this ameliorates the selection of the closest points pairs and thus the subsequent steps. Third, and most crucial, in standard ICP the identified scaling matrix $S$ is discarded and $R = UV^T$ constitutes the final output, whereas in our method $S$ contains the sought answer and constitutes the main final output of our method.

**Interpretation of extracted data.** Several numerical values in Eq (4) have a direct, clinically relevant interpretation. Unit vectors $u_i$ and $v_i$ (with $i$ in 0,1,2) represent the directions of maximal, median, and a minimal expansion in the reference frames of $M_0$ and $M_1$ respectively. In all our experiments except one, the longitudinal axis (orthogonal to the mid-sagittal plane, and represented by the X-axis in our dataset) in the respective reference systems approximatively matched the direction of maximal expansion $u_0$, with it matching the direction of median expansion $u_1$ in one case.

The primary data which is returned by the system is the sought overall longitudinal enlargement factor, which is the scalar value $\sigma_i$ linked to expansion direction that is most similar to the X-axis (reported in bold in Table 1). This value is a dimensionless value which is reported to the user and aggregates all the 3D measurements used by the ICP.

**Table 1. Experimental results with several patients.**

| Clinical case | | | Affine alignment | | | | | Rigid alignment | |
|---|---|---|---|---|---|---|---|---|---|
| | | | Singular values | | | Alignment error | | Alignment error | |
| Treatment | Age (years) | Delta age (months) | $\sigma_1$ | $\sigma_2$ | $\sigma_3$ | average (mm) | variance (mm) | average (mm) | variance (mm) |
| Hyrax | 15 | 32 | **1.106** | 1.017 | 0.916 | 0.653 | 0.595 | 1.301 | 1.227 |
| | 12 | 35 | **1.093** | 1.017 | 0.888 | 0.696 | 0.431 | 1.165 | 0.683 |
| | 15 | 36 | **1.073** | 0.982 | 0.816 | 0.657 | 0.43 | 1.147 | 0.699 |
| | 10.5 | 12 | **1.173** | 1.084 | 0.993 | 0.712 | 0.444 | 1.154 | 0.645 |
| | 9 | 36 | **1.066** | 1.004 | 0.889 | 0.52 | 0.345 | 0.76 | 0.431 |
| | 14.5 | 35 | **1.188** | 0.994 | 0.919 | 0.531 | 0.337 | 1.379 | 1.086 |
| | 10.5 | 22 | **1.125** | 1.05 | 1.003 | 0.59 | 0.467 | 1.062 | 0.818 |
| | 11.5 | 22 | **1.031** | 1.02 | 0.938 | 0.441 | 0.228 | 0.775 | 0.418 |
| | 13 | 8 | **1.107** | 1.091 | 0.897 | 0.528 | 0.186 | 1.068 | 0.56 |
| | 16.6 | 35 | **1.068** | 1.005 | 0.842 | 0.664 | 0.631 | 1.082 | 0.687 |
| average | 12.76 | 27.3 | **1.103** | 1.026 | 0.91 | 0.599 | 0.409 | 1.089 | 0.725 |
| Removable | 9.5 | 46 | 1.227 | **1.067** | 0.986 | 0.652 | 0.463 | 0.944 | 0.842 |
| | 7 | 27 | **1.136** | 1.061 | 0.836 | 0.695 | 0.517 | 0.917 | 0.323 |
| | 9 | 7 | **1.036** | 1.025 | 0.97 | 0.4 | 0.261 | 0.5 | 0.24 |
| | 6.5 | 14 | **1.051** | 0.999 | 0.926 | 0.649 | 0.719 | 0.77 | 0.67 |
| | 9 | 10 | **1.084** | 1.002 | 0.902 | 0.611 | 0.485 | 0.638 | 0.596 |
| | 9.5 | 25 | **1.026** | 0.994 | 0.957 | 0.579 | 0.559 | 0.584 | 0.505 |
| | 10 | 10 | **1.098** | 1.003 | 0.997 | 0.392 | 0.191 | 0.731 | .341 |
| | 9 | 42 | **1.115** | 1.014 | 0.923 | 0.583 | 0.313 | 0.876 | 0.367 |
| | 9 | 43 | **1.044** | 0.996 | 0.945 | 0.467 | 0.306 | 0.721 | 0.454 |
| | 9.5 | 19 | **1.021** | 1.014 | 0.949 | 0.663 | 1.296 | 0.719 | 1.184 |
| average | 8.8 | 24.3 | **1.084** | 1.017 | 0.939 | 0.569 | 0.511 | 0.74 | 0.552 |
| average (all) | 10.78 | 25.8 | **1.093** | 1.022 | 0.925 | 0.584 | 0.46 | 0.915 | 0.639 |

For each case, we report the age of the subject, the time passed between the two scans, and the extracted enlargement factor (in bold), which corresponds to the singular value extracted from the alignment matrix which corresponds to the enlargement axis most similar to the longitudinal direction. For completeness, we also report the other two singular values. The table also shows the residual alignment errors that are obtained using the proposed affine alignment, and the significantly larger errors obtained with the traditional rigid alignment.

To communicate visually the information about the detected directions of expansion, we have integrated into MeshLab the possibility to display the orthogonal scaling directions of the transformation applied to the aligned mesh (i.e., the vectors $u_0$, $u_1$, $u_2$). The other values in Eq (4) are not useful to our analysis and contain no reliable information. The product $\sigma_0 \cdot \sigma_1 \cdot \sigma_2$ (i.e., the modulus of the determinant of $A$) has also a direct geometric interpretation, representing the scaling factor of the total volume, but our experimental data indicates that this datum cannot be reliably used.

Several numerical values in Eq (4) have a direct clinically relevant interpretation. The single scalar value $\sigma_0$ directly represents the sought relative expansion coefficient. It is a dimensionless value that is reported to the user and it implicitly aggregates all the 3D measurements used by the ICP. Unit vectors $u_0$ and $v_0$ represent the directions of maximal expansion in the reference frames of $M_0$ and $M_1$ respectively. In our experiments, these often matched the longitudinal axes (orthogonal to the mid-sagittal plane) in the respective reference systems. The other values in Eq (4) are not useful to our analysis and contain no reliable information. The product $\sigma_0 \cdot \sigma_1 \cdot \sigma_2$ (i.e., the modulus of the determinant of $A$) has also a direct geometric interpretation, representing the scaling factor of the total volume, but our experimental data indicates that this cannot be reliably used.

## Software implementation

We implemented our method as an extension of the open-source MeshLab [14] 3D processing system. This MeshLab extension is offered as a publicly available pull request on the Open-Source GitHub repository of MeshLab and serves as a reference implementation of our proposed method.

In our new extended version, the new functionality is made available to the operator in the form of a new setting for the alignment of two given polygonal meshes, along with the possibility to display the orthogonal scaling directions of the transformation applied to the aligned mesh, i.e., the vectors $u_0$, $u_1$, $u_2$), and the corresponding scaling coefficients (Fig 2).

We opted for this solution so that our method is integrated into a complete interactive 3D suite, which can be conveniently leveraged to perform the preliminary cropping and coarse alignment of the input meshes (phases 0 and 1 of our method) before computing the affine non-rigid alignment and extracting the relevant data.

**Implementation details.** Internally, the linear least-squares system solution and the computation of the SVD are implemented using the Eigen C++ library [15]. The closest point identification is implemented via a regular grid spatial indexing structure using the VCG library [16].

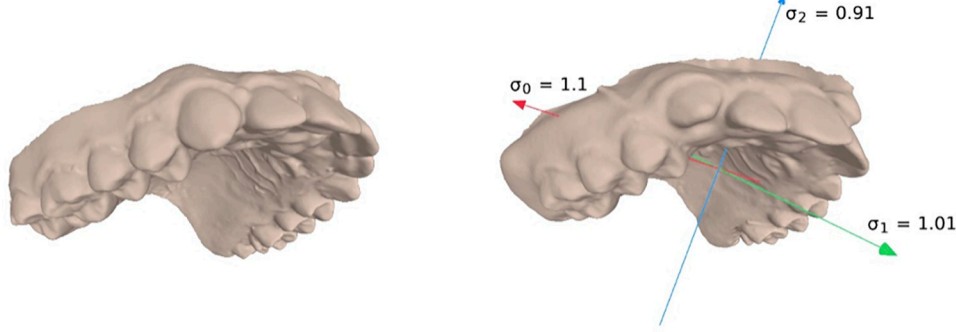

**Fig 2.** Target shape (left, post-treatment) and pre-treatment deformed shape with expansion directions and coefficients extracted from the affine matching matrix.

## Experimental setup

Initially, 125 pre- and post-treatment DM were randomly selected from the archive of the University of *[redacted]*, and 40 with the following characteristics were selected for the study:

- Aged between 10 ± 5 years, both sexes.

- At least two DM with a time span of 21 ± 10 months taken before and after ($M_0$, $M_1$) the following treatments: Rapid maxillary expander (Hyrax) or NiTi self expander (Leaf), equally distributed between the two groups according to the protocol of Chaconas et al. 1982 [17] and Lanteri et al. 2016 [18].

- No syndromes or previous history of traumatic cranio-facial injuries or chronic disease.

- Sufficient mesh quality and correct display of palatal anatomy and palatal rugae according to Almeida et al. 1995 [5], with at least 40–50.000 triangular faces.

- To avoid scanning errors related to the operator's experience in the scanning process only 3D models obtained from extraoral laboratory scanners were considered.

For the minor patients all the patients' legal representatives were informed about the study and signed written informed consent prior to the realization of the following research. The usage of anonymized data follows the Helsinki declaration. IRB was received from 1–2021, [REDACTED]; 07.01.2021

## Pre-alignment coarse aligning procedures

The selected pairs of DM ($M_0$ and $M_1$) were then imported into an open-source system for processing and editing 3D triangular meshes MeshLab [14]. The point-based alignment function was used for a fast and reliable pre-alignment of the $M_0$ and $M_1$ using palatal rugae as a reference. A set of at least 10 points were arbitrarily chosen by an expert operator (MF). After a first rough alignment, the ICP algorithm was used to refine the affine matching between the two meshes.

## Results

As can be seen from Table 1, non-rigid alignment provided a realistic quantification of palatal expansion of 9.3% (SD 4.5%) in a time span of 24 months.

Separately, for the Hyrax group a total mean expansion of 10.3% (SD 4.6%) was observed over an average time span of 27.3 months (mean age 12.5 years).

For NiTi automatic expander a total mean expansion of 8.4% (SD 3.7%) was observed with an average time span of 24 months (mean age 8.5 years).

We also compare the alignment error resulting from our proposed affine matching with the one resulting from the traditional rigid matching (four last columns in Table 1). Alignment errors are computed with MeshLab as the average and variance of the residual distance between the two surfaces, after alignment. We observe the affine alignment error to be significantly and consistently lower. The difference confirms our conjecture: the actual physical deformation of the maxilla is much better approximated by an affine transformation, which can also include anisotropic scaling in arbitrary directions, than by a rigid transformation, which only accounts for reorientation and translation. We remark, however, that our objective is not to improve on the alignment error, but to robustly extract useful data from the alignment transformation itself, and offer it to the user as a concise, meaningful, and descriptive characterization of the observed deformation.

## Assessing robustness and precision

We also perform a separate experiment to assess the precision of our method, specifically in terms of its robustness to noise and inaccuracies in the manual landmark selection that the user must provide to construct the initial coarse alignment.

Our setup for this experiment is as follows. We use an arbitrary pair of scans (first raw of Table 1: subject aged 15, range-scans captured 32 months apart). We pick 10 landmark pairs for the initial manual alignment, as by our protocol. We artificially perturb each landmark by displacing its 3D position; the displacement is obtained by adding a Gaussian distributed random error with 0 mean and $k$ standard deviation, simulates inaccuracies by the operator in the selection of the correspondences; we then proceed, as normal, by automatically refining the alignment with our modified ICP procedure, extracting and recording the final expansion coefficients.

We perform 30 runs of this experiment with increasing values of error value $k$. We use $k = 0, 1, 2, 4, 8$ and 16 millimeters, repeating five runs for each of the six values of $k$.

The numerical results of all runs are graphed in Fig 3. The data shows that the extracted expansion coefficients are only minimally affected by inaccuracies in manual point selection. In the last sequence of runs, for example, the artificial simulated errors (16 mm) unrealistically exceed the combined widths of entire teeth, yet the extracted expansion coefficients range only between 1.010 and 1.011; smaller, more realistic injected errors results in only subpercentage differences in detected expansion coefficient; (note that, even when $k = 0$, we still observe some negligible but non-zero variation of extracted coefficients; this is due to the randomized nature of the ICP algorithm).

The experiment suggests that our method produces consistent expansion coefficients, which reflect the input range-scanned data, rather than the particular set of points selected to initialize the ICP. This is because the manual selection only serves as an initialization for the IPC, which, irrespectively, converges to similar or identical solutions.

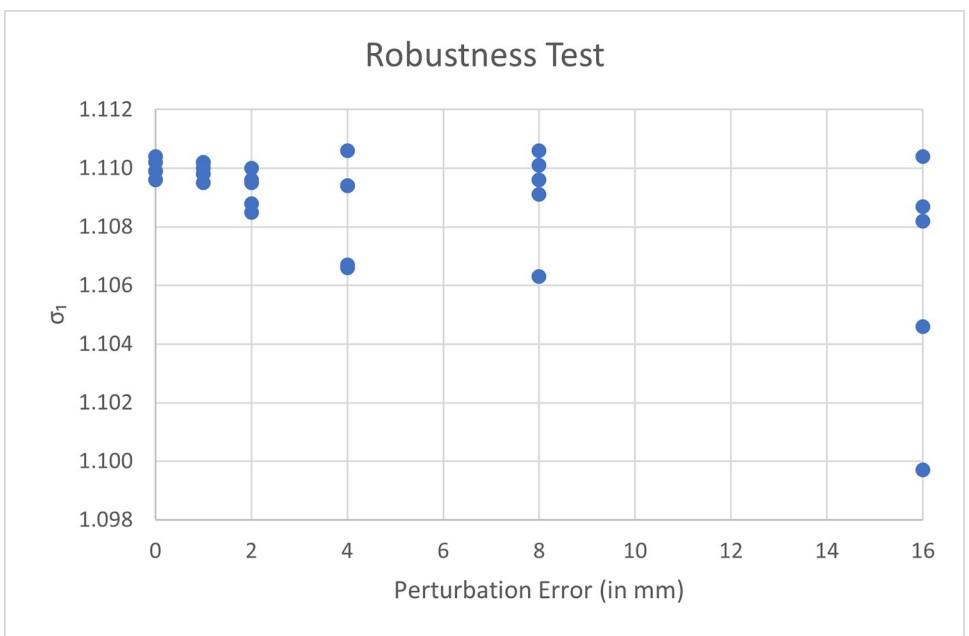

**Fig 3. Expansion coefficient values extracted by repeated trials with increasing perturbation noise applied to the initial landmarks used for rough alignment.** Note that, when no perturbation is applied, the final coefficients still differ slightly across trials due to the randomness of the ICP sampling scheme adopted by MeshLab.

## Discussion

In the present paper, a new non-rigid alignment superimposition original algorithm was applied on a small sample of orthodontic patients treated with two different types of palatal expanders in order to overcome the limitations imposed by rigid alignments [18] and provide a more meaningful assessment of size and shape changes following a treatment.

Originally the shape of an object is defined, technically, as all the geometric features of an object except for its size, position, and orientation [19]. Based on this interpretation, being the shape the main source of information for analyzing images, we obviously need to discount information on the size, position, and orientation of a biological object. Based on these assumptions, diagrams showing individual shapes and visualizations that display a combination of two or more shapes to show the differences between them may look very different because we can display the shapes of objects without worrying about their size, position, and orientation. These last can support potentially misleading clinical interpretations if clinicians neglect this part of the picture.

Unfortunately, visualization through landmark displacements and through graphs based on deformation, such as transformation grids and warped 3D surfaces, may hide the vectors of the dimensional changes of the biological structure investigated for example subjected to a specific mechanical or surgical treatment. Moreover, whereas a shape corresponds to a single point in shape space or shape tangent space, a shape change is the movement from the point representing the starting point to the point representing the target shape. This means that it is a vector that has a direction and a magnitude expressed only in percentage [20].

In other words, shape changes always need to be visualized in conjunction with a shape. In order to interpret the change in shape, we need to understand the relative displacements of landmarks in the context of their overall arrangement. Thus, shape changes are only interpretable in the context of the structure for which they were found and in conjunction with the shape of that structure.

GMA is a useful tool in clinicians' hands to qualitatively study the changes induced by different mechanical or surgical treatment options. It allows the visualization of shape changes as images with different color grades providing information regarding biometric differences in its anatomical contour [21].

Moreover, shape changes and eventually derived linear measurements are strictly dependent on the reference plane [22]. Used in this manner, GMA visualization of shape changes provides powerful means for communicating complex results in an intuitively and appealingly, but not enough for clinical purposes.

In our study, hard tissue landmarks were semiautomatically collected on digital dental 3D models. All subsequent measurements and calculations were automatically performed by an original computerized mathematical equation. The present protocol, thus, allowed the analysis of digital models with a method error only limited to the repeatability of landmark identification. With this approach, three-dimensional anatomical changes obtained with medical and dental treatments can also be quantitatively analyzed. Indeed, in cases of clinical evaluation, we consider the renounce to any size information between superimposed objects a limitation. Instead, sources, directions, and measures with the non-rigid alignment approach are, highlighted and they become a useful tool to understand, for example, the entity of movement needed during an orthodontic treatment.

The measurement protocol and the equations used in the present investigation appear to be practical tools for the quantitative description of human hard tissue palate subjected to expansion procedures.

The same mathematical description could be extended to all the craniofacial area and to other anatomical structures. Digitized data can also enter in any kind of mathematical modeling, thus offering, for instance, new possibilities with artificial intelligence and deep learning algorithms to the pretreatment computerized previews of the expected result.

## Limitations and future work

Our current solution relies on the main assumption that an affine transformation can describe, with sufficient approximation, the physical deformation that occurred in a specific area over the entire anatomical structure. A limitation that must be taken into account is the scanning error. We did not measure the accuracy of our 3D models as we considered it out of the scopes of the study, but it should be taken into account that an error exists when using a laboratory scanner. According to recent studies the accuracy error ranges from 21.3 μm to 33.8 μm [23].

While the scanning methodology might have introduced this error in our study, we are confident that the use of a digital scanner has a key role in the automatization of the process, direct in vivo measurements could also be a source of error and the use of IOS is generally considered less precise than the laboratory scanner by the scientific community [24]. In order to extend the presented approach to any other scenario, it would be necessary to further generalize the class of non-rigid transformation beyond simple affine transformations. Unfortunately, the problem of nonlinear, non-rigid alignment is known to be extremely difficult to solve in a robust, reliable way [25], and even more so in presence of incomplete, high-resolution 3D data. Furthermore clinical setting differentiations could affect the results achieved, for example it is known that the dental elements extraction could affect the reliability and position of the rugae, with we used for the coarse initial alignment. In the same way we suggest caution when in presence of MARPE and SARPE expansion techniques. Technical difficulties aside, we think that the core idea behind our solution can be profitably extended in many other different domains, and namely, that the (non-rigid) alignment between 3D data is not just a preliminary necessity for the analysis of 3D images, but it contains reliable clinically relevant information which can be easily extracted. A future implementation of our study will propose the integration with CT data for affine alignment, however under the current ethical consideration, to date, the use of before and after radiologic exams is discouraged under normality conditions [26].

## Supporting information

**S1 Data.**
(WEBLOC)

## Author Contributions

**Conceptualization:** Andrea Maggiordomo, Marco Farronato, Gianluca Tartaglia, Marco Tarini.

**Data curation:** Andrea Maggiordomo, Marco Farronato.

**Formal analysis:** Andrea Maggiordomo, Marco Farronato.

**Methodology:** Gianluca Tartaglia.

**Software:** Andrea Maggiordomo, Marco Tarini.

**Supervision:** Gianluca Tartaglia.

**Writing – original draft:** Andrea Maggiordomo, Marco Farronato.

**Writing – review & editing:** Marco Tarini.

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
