## [Decision Letter · Decision Letter 0]

14 Jun 2022

PONE-D-21-38690A method based on 3D affine alignment for the quantification of palatal expansionPLOS ONE

Dear Dr. Farronato,

Thank you for submitting your manuscript to PLOS ONE. Firstly, we would like to apologize for the delay in processing your manuscript. It has been exceptionally difficult to secure reviewers to evaluate your study. We have now received two completed reviews, which are available below.

After careful consideration, we feel that it has merit but does not fully meet PLOS ONE’s publication criteria as it currently stands. Especially Reviewer #1 raised several scientific concerns about the current manuscript. Therefore, we invite you to submit a revised version of the manuscript that addresses the points raised during the review process. 

We look forward to receiving your revised manuscript.

Kind regards,

Thomas Tischer

Staff Editor

PLOS ONE

**Journal requirements:**

PLOS ONE does not provide copy-editing, please carefully ensure the use of standard English language and grammar throughout the manuscript.Please ensure that all references adhere to the PLOS ONE style guide: https://journals.plos.org/plosone/s/submission-guidelines#loc-references  

**Additional Editor Comments:**

Please discuss the limitations of your current approach based on the points mentioned by Reviewer #2.  

Reviewers' comments:

Reviewer's Responses to Questions

**Comments to the Author**

1. Is the manuscript technically sound, and do the data support the conclusions?

Reviewer #1: No

Reviewer #2: Partly

2. Has the statistical analysis been performed appropriately and rigorously? 

Reviewer #1: N/A

Reviewer #2: Yes

3. Have the authors made all data underlying the findings in their manuscript fully available?

Reviewer #1: Yes

Reviewer #2: No

4. Is the manuscript presented in an intelligible fashion and written in standard English?

Reviewer #1: Yes

Reviewer #2: No

5. Review Comments to the Author

Reviewer #1: Dear editor, dear colleagues,

I am grateful for the opportunity to read and comment on this paper. This paper describes an interesting new approach to measure maxillary changes after palatal expansion. We need more research about this topic, and I was glad to find researchers working on it. Furthermore, the authors use open-source software and intent to publish all their data. Therefore, open-source software should be supported since the method will be easily accessible for researchers and clinicians worldwide. However, I recommend publication with major changes that I would like to explain below.

The authors used 20 pairs of plaster casts taken before and after treatment with two different RME appliances to evaluate a new approach to measuring changes and treatment effects in the palatal vault. First, the authors describe the problem with commonly used ICP-based procedures using rigid alignment procedures: these procedures do not consider that the palate's shape changes significantly during the orthodontic treatment. Therefore, the authors propose a non-rigid approach instead. However, even this non-rigid approach depends on the 3D models being placed somewhat correctly in the 3D grid. Therefore, the starting point is a coarse alignment according to manually chosen reference points close to the first pair of rugae and the palatal midline. Then an ICP algorithm is applied, further refining the positioning. Finally, in the third step, the change in shape is calculated. This approach reminds me somehow of the RFD superimposition method, but the third step is a different algorithm.

First things first. This paper needs to be clear about what it is. Therefore, the authors must define the primary research question and the primary outcome measure. The aim and the primary research question are tightly connected, and they can typically be found in the last section of the introduction. The authors write: "The present research aims to propose a new method for non-rigid alignment of 3D digital reproductions of dental arches and related structures to precisely evaluate dimensions and shape changes between two meshes taken, for example, at different time frames."

Maybe this is more what the authors want, to propose the new method. However, the aim of this paper would rather be to compare expansion measurements according to the new method with measurements retrieved with the rigid method.

Here we also see what my main concern with this paper is. It is not a validation study for one reason: We do not learn anything about accuracy & precision. To evaluate precision, one would expect intra- and inter-examiner reliability results. To assess the accuracy, one would need to create a specimen in 3D software, e.g. to simulate palatal expansion. In that way, the 100% correct result would be known to the authors, and they could test the results of their method against the truth.

In the current version of this paper, we only learn that the average alignment error of the non-rigid approach is minor compared to the rigid method. But what does this tell us? Maybe the difference between the two models should be according to the rigid method. However, since we do not know the actual value (see above, no specimen), we cannot be sure that a lower average alignment error equals a better matching result.

I assume that the average alignment error is the same as the root-mean-square (RMS) frequently used in trials. However, I'm afraid I disagree that RMS is suitable for this evaluation. The RMS is calculated on a number of randomly distributed reference points. Thus, the RMS does not take the whole geometry into account. If the actual value was known, one could conduct the superimposition according to both methods and then calculate a distance analysis. Distance analysis usually calculates differences on all points or triangles of the mesh.

I understand that this work is finished and that my suggestions imply that this project needs to go back to the workbench. Simulating the expansion with a 3D software and creating a specimen is possibly too much. However, the authors could at least conduct further measurements with the same operator and some other operators. Then, we would learn more about intra- and inter-examiner reliability. Accordingly, the results section would have a bit more material. Unfortunately, the results are way too short.

Yet some other important aspects:

The authors describe that the reference points for the coarse alignment are around the first pair of rugae. In the literature, we read that the median point of the third pair of rugae is regarded as most but not 100% stable. The authors are right in choosing the first pair of rugae since these cases are treated with palatal expanders, and the third pair of rugae might be affected in the vertical dimension when the palate "flattens" during expansion. However, for extraction cases, the first pair of rugae is heavily influenced by orthodontic treatment. The authors need to mention this more clearly in the discussion: the more we know about the conducted treatment, the better the superimposition gets because we can choose the suitable reference structures. Unfortunately, we are still far from a one-click solution that fits all situations. Only this is an exciting finding worth being published.

What is the clinical relevance, and how does your finding correspond to the traditional measures. For example, the author could add traditional measures to their data such as intermolar distance, inter canine distance, arch length, and maxilla depth. In that way, your readers can easily relate your results to what they are used to seeing when discussing palatal expansion.

Since this topic is not easy to understand, I would appreciate more images. When you use figures containing multiple images, you need to name the images with letters A, B and C to refer to single images in the text.

Talking about images, Figure 1 needs more explanation. I assume that all three images are from the same case. From the image to the right to judge, the 3D model literally is stretched with the affine method. Is this stretched version only used for superimposition purposes to calculate the correct position of the original untreated maxilla? I hope so. A bit more explanation would be great.

Some minor comments

Keeping in mind that I am not an English native speaker, I do not feel familiar with the word "mold" or "3D cast". Cast derives from the production process of casting. Thus, I would rather use "plaster casts" instead of mold and "3D models" instead of 3D cast.

Reviewer #2: The quality of the English is not at an appropriate level for the journal.

There are several typos. Some sentences are very long and make it difficult to understand. The article is not flowing.

Although the topic is of great interest, there are bias in the research methodology. No account was taken of errors due to impression taking or the systematic error of the scanners used. Furthermore, the study is not carried out on patients but on models. For this reason, the clinical application of the study is limited.

6. PLOS authors have the option to publish the peer review history of their article (what does this mean?). If published, this will include your full peer review and any attached files.

Reviewer #1: **Yes: **Niels Ganzer

Reviewer #2: No

---

## [Author Response · Author response to Decision Letter 0]

7 Sep 2022

A method based on 3D affine alignment for the quantification of palatal expansion

Response to reviewers

Journal requirements and editor comments

We revised file names, corrected the formatting of authors and affiliations, and updated the references style.

We modified our statement as follows:

For the minor patients all the patients’ legal representatives were informed about the study and signed written informed consent prior to the realization of the following research. The usage of anonymized data follows the Helsinki declaration. IRB was received from 1-2021, [REDACTED]; 07.01.2021

We moved our ethic statement in the appropriate section.

PLOS ONE does not provide copy-editing, please carefully ensure the use of standard English language and grammar throughout the manuscript.

We checked the spelling and rewrote some sentences.

Please ensure that all references adhere to the PLOS ONE style guide: https://journals.plos.org/plosone/s/submission-guidelines#loc-references

We corrected the references.

Please discuss the limitations of your current approach based on the points mentioned by Reviewer #2.

Dear Editor, we revised our manuscript to answer the questions raised in the first round of reviews, and clarify some fundamental differences between rigid alignment and our method based on geometric information extracted from the affine alignment. We answered Reviewer #2 questions and we added data retrieved from the literature, kindly let us know if this is what you meant.

Reviewer #1

This paper describes an interesting new approach to measure maxillary changes after palatal expansion. We need more research about this topic, and I was glad to find researchers working on it. Furthermore, the authors use open-source software and intent to publish all their data. Therefore, open-source software should be supported since the method will be easily accessible for researchers and clinicians worldwide. However, I recommend publication with major changes that I would like to explain below.

Dear Reviewer, we are very thankful for your kind premise, this multidisciplinary research was dictated by the clinical urge to provide new solutions for the automatic analysis, a fundamental method for different orthodontic procedures, pre and post palatal expansion among those. We are grateful for your deep and thorough analysis, if and when preliminary scientific soundness would be achieved we believe open source release of the algorithms developed stands in favor of progress and research freedom, allowing other researched to further implement or to replicate the results achieved. Below you will find our comments and improvements based on your precious revision.

The authors used 20 pairs of plaster casts taken before and after treatment with two different RME appliances to evaluate a new approach to measuring changes and treatment effects in the palatal vault. First, the authors describe the problem with commonly used ICP-based procedures using rigid alignment procedures: these procedures do not consider that the palate's shape changes significantly during the orthodontic treatment. Therefore, the authors propose a non-rigid approach instead. However, even this non-rigid approach depends on the 3D models being placed somewhat correctly in the 3D grid. Therefore, the starting point is a coarse alignment according to manually chosen reference points close to the first pair of rugae and the palatal midline. Then an ICP algorithm is applied, further refining the positioning. Finally, in the third step, the change in shape is calculated. This approach reminds me somehow of the RFD superimposition method, but the third step is a different algorithm.

Dear reviewer, thanks for suggesting the RFD method, we added a section with a comparison with the previously existing method in the introduction.

The initial coarse alignment is a manual procedure that is determined by the manual selection of few landmarks, but is a necessary step when aligning 3D surfaces with all ICP-based methods (rigid or affine), which are inherently local and otherwise are not guaranteed to converge to the “right” alignment. However, our method is not particularly sensitive to errors in this initial coarse alignment, whereas methods based on rigid alignment require careful selection of the landmarks by an expert operator.

In the Results section, we now report the results of an experiment that we conducted to show that our measure is quite robust to noise and inaccuracies in the landmark selection of the initial rough alignment.

First things first. This paper needs to be clear about what it is. Therefore, the authors must define the primary research question and the primary outcome measure. The aim and the primary research question are tightly connected, and they can typically be found in the last section of the introduction. The authors write: "The present research aims to propose a new method for non-rigid alignment of 3D digital reproductions of dental arches and related structures to precisely evaluate dimensions and shape changes between two meshes taken, for example, at different time frames."

Maybe this is more what the authors want, to propose the new method. However, the aim of this paper would rather be to compare expansion measurements according to the new method with measurements retrieved with the rigid method.

Dear reviewer, thanks for your suggestion, we agree with you, we added a specific section in the introduction, and, in general, we rewrote some sentences. This revision clarifies that the scope of our paper is quantifying the deformation resulting from palatal expansion treatments with a measure that is geometrically derived from the deformation itself, and is novel in the dentistry field.

Here we also see what my main concern with this paper is. It is not a validation study for one reason: We do not learn anything about accuracy & precision. To evaluate precision, one would expect intra- and inter-examiner reliability results. To assess the accuracy, one would need to create a specimen in 3D software, e.g. to simulate palatal expansion. In that way, the 100% correct result would be known to the authors, and they could test the results of their method against the truth.

In the current version of this paper, we only learn that the average alignment error of the non-rigid approach is minor compared to the rigid method. But what does this tell us? Maybe the difference between the two models should be according to the rigid method. However, since we do not know the actual value (see above, no specimen), we cannot be sure that a lower average alignment error equals a better matching result.

Dear reviewer, thanks for your kind observation, here comes the dilemma of palatal expansion, in our opinion it is not clear how to establish a “ground truth” of a body which moves completely. In our work, we introduce an automatic method which 1) requires only minimal human intervention to produce an approximation of the non-rigid movement of thousands of points (the samples/vertices of the digital 3D scans) and 2) it is geometrically derived, hence intrinsically meaningful as it captures and quantifies the geometry of the deformation itself, rather than simple distances between points in the two scanned surfaces.

In this revised version we have expanded the results section with a discussion and clarification on the meaning of reporting and comparing the surface distances (average and variance) with affine and rigid alignments. We clarify that this shows how the affine matching can exploit the extra degrees of freedom of the linear non-rigid deformation to produce a much better approximation of the target surface, giving meaning to the measures we then extract from the deformation itself.

Moreover, in this revised version, we have included an experiment to validate our approach under random perturbations of the initial rough alignment.

I assume that the average alignment error is the same as the root-mean-square (RMS) frequently used in trials. However, I'm afraid I disagree that RMS is suitable for this evaluation. The RMS is calculated on a number of randomly distributed reference points. Thus, the RMS does not take the whole geometry into account. If the actual value was known, one could conduct the superimposition according to both methods and then calculate a distance analysis. Distance analysis usually calculates differences on all points or triangles of the mesh.

I understand that this work is finished and that my suggestions imply that this project needs to go back to the workbench. Simulating the expansion with a 3D software and creating a specimen is possibly too much. However, the authors could at least conduct further measurements with the same operator and some other operators. Then, we would learn more about intra- and inter-examiner reliability. Accordingly, the results section would have a bit more material. Unfortunately, the results are way too short.

Dear reviewer we performed the following modifications:

- Added discussion on rigid vs affine matching

- Added robustness test

The authors describe that the reference points for the coarse alignment are around the first pair of rugae. In the literature, we read that the median point of the third pair of rugae is regarded as most but not 100% stable. The authors are right in choosing the first pair of rugae since these cases are treated with palatal expanders, and the third pair of rugae might be affected in the vertical dimension when the palate "flattens" during expansion. However, for extraction cases, the first pair of rugae is heavily influenced by orthodontic treatment. The authors need to mention this more clearly in the discussion: the more we know about the conducted treatment, the better the superimposition gets because we can choose the suitable reference structures. Unfortunately, we are still far from a one-click solution that fits all situations. Only this is an exciting finding worth being published.

Dear reviewer, thanks, we appreciate your kind comment, there could be a lot more limitations to be added, and we think this preliminary result could be a starting point. We added the extraction considerations and other limitations as the SARPE or MARPE keeping in mind that there could be more to add in the future.

What is the clinical relevance, and how does your finding correspond to the traditional measures. For example, the author could add traditional measures to their data such as intermolar distance, inter canine distance, arch length, and maxilla depth. In that way, your readers can easily relate your results to what they are used to seeing when discussing palatal expansion.

Since this topic is not easy to understand, I would appreciate more images. When you use figures containing multiple images, you need to name the images with letters A, B and C to refer to single images in the text.

Talking about images, Figure 1 needs more explanation. I assume that all three images are from the same case. From the image to the right to judge, the 3D model literally is stretched with the affine method. Is this stretched version only used for superimposition purposes to calculate the correct position of the original untreated maxilla? I hope so. A bit more explanation would be great.

Dear reviewer, we added another image and modified the captions, kindly let us know if it is more clear now: Comparison of affine (c) and rigid (b) registration errors (increasing scale towards red) when superimposed to the post-treatment model (a). Affine registration accurately models the palatal deformation producing a noticeably lower alignment error of the scans pre and post-treatment (showing uniform blue coloration).

Some minor comments

Keeping in mind that I am not an English native speaker, I do not feel familiar with the word "mold" or "3D cast". Cast derives from the production process of casting. Thus, I would rather use "plaster casts" instead of mold and "3D models" instead of 3D cast.

Dear reviewer, thanks, we corrected the typos and, generally, we made sure to use standard terminology.

Reviewer #2

The quality of the English is not at an appropriate level for the journal.

There are several typos. Some sentences are very long and make it difficult to understand. The article is not flowing.

Dear Reviewer, thanks for giving us your opinion on our research, we revised the use of the English language with the help of a native speaker.

Although the topic is of great interest, there are bias in the research methodology. No account was taken of errors due to impression taking or the systematic error of the scanners used. 

Dear reviewer, thanks for this comment, based on a systematic review we added the scanner average error, this is now included in the limitation section. Generally the IOS scanning overall error ranges from 30.4 μm to 98.4 μm (Mangano, F. G., Admakin, O., Bonacina, M., Lerner, H., Rutkunas, V., & Mangano, C. (2020). Trueness of 12 intraoral scanners in the full-arch implant impression: a comparative in vitro study. BMC oral health, 20(1), 263. https://doi.org/10.1186/s12903-020-01254-9) while the laboratory scanner range from 21.3 μm to 33.8 μm. (Ebeid, K., Nouh, I., Ashraf, Y., & Cesar, P. F. (2022). Accuracy of different laboratory scanners for scanning of implant-supported full arch fixed prosthesis. Journal of esthetic and restorative dentistry : official publication of the American Academy of Esthetic Dentistry ... [et al.], 10.1111/jerd.12918. Advance online publication. https://doi.org/10.1111/jerd.12918) We decided not to perform any evaluation of the scanning error, due to this being a well described topic in the literature and being out of the scopes of the study, although we agree it is an important question and we added it to the limitation section.

Furthermore, the study is not carried out on patients but on models. For this reason, the clinical application of the study is limited.

We are not sure we understood this comment, the automatic digital analysis of the palatal structure/changes is usually carried on digital models. Intra-oral direct measurements to the author’s knowledge cannot be taken automatically.

---

## [Editor Report · Decision Letter 1]

15 Nov 2022

A method based on 3D affine alignment for the quantification of palatal expansion

PONE-D-21-38690R1

Dear Dr. Farronato,

We’re pleased to inform you that your manuscript has been judged scientifically suitable for publication and will be formally accepted for publication once it meets all outstanding technical requirements.

Kind regards,

Martina Ferrillo

Academic Editor

PLOS ONE

---

## [Editor Report · Acceptance letter]

21 Nov 2022

PONE-D-21-38690R1 

A method based on 3D affine alignment for the quantification of palatal expansion 

Dear Dr. Farronato:

I'm pleased to inform you that your manuscript has been deemed suitable for publication in PLOS ONE. Congratulations! Your manuscript is now with our production department. 

Kind regards, 

on behalf of

Dr. Martina Ferrillo 

Academic Editor

PLOS ONE